# Can Simple Metrics Identify the Process(es) Driving Extreme Precipitation?

**Leif M. Swenson** 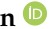

Atmospheric Science Program, Department of Land, Air and Water Resources , University of California, Davis, CA 95616, USA; lmswenson@ucdavis.edu

**Abstract:** This work seeks an automatic algorithm to determine the primary meteorological cause(s) of individual extreme precipitation events. Such determinations have been made before, but required a by-hand analysis of each separate event. This is very time-consuming and the field would benefit from an automatic process. This is especially relevant when comparing different datasets to determine which ones most closely hew towards reality. This paper tests three simple metrics over the continental United States using the European Center for Medium-Range Weather Forecasting's (ECMWF) atmospheric reanalysis (ERA5). The metrics tested measure and compare the strength of three meteorological processes associated with extreme precipitation: fronts, convection, and cyclones. A multivariate statistical technique as well as individual case studies show evidence that the three meteorological processes of interest cannot be isolated from one another using these simple physical metrics. This shows the difficulty in finding "pure" cases of these precipitation-generating processes and suggests approaching these processes with an eye toward mixed-type events.

**Keywords:** extreme precipitation; climate; midlatitude weather; reanalysis

## 1. Introduction

This study presents an approach to quantifying the relative importance of various meteorological processes to the generation of specific extreme precipitation (PEx) events. This work builds on physically based existing metrics with a new quantitative approach to comparing the relative strengths of the underlying processes being measured. The motivation is the desire to find an automated algorithm analogous to the results described in Kunkel [1]. The Kunkel [1] method uses a manual examination of each event to identify a single meteorological process responsible for each continuous area with precipitation $> 12.5 \frac{mm}{day}$. This is very time-consuming and the field would benefit from an automated identification process. However, the results presented here indicate that these physically based metrics are unable to unambiguously distinguish between various meteorological processes across areas with continuous areas of precipitation as described in Kunkel [1]. The three meteorological processes considered are fronts, convection, and midlatitude cyclones. To assess the ability to distinguish between these meteorological processes of interest, the results of this work are compared to several individual cases from Kunkel [1] as well as the seasonal trends from Kunkel [1] and Dowdy [2]. The presented results suggest that these meteorological processes are not distinguishable at the grid-point level. Section 2 provides a background of previous detection and classification algorithms. Section 3 discusses the methods used in this work. Section 4 presents the results, which are discussed in more detail in Section 5, and Section 6 presents conclusions.

## 2. Background

The importance of meteorological processes that create extreme precipitation (e.g., fronts, convection, and extratropical cyclones) is known to vary seasonally and spatially [1]. There have been numerous case studies where these meteorological processes (fronts,

convection, and extratropical cyclones) have been identified for individual extreme events. That procedure is undertaken by hand and can be quite labor-intensive. This labor can become a prohibitive barrier to creating a climatology of the driving meteorological process behind precipitation events. There has been at least one notable attempt to do this from observed data [1], which required hand analysis of thousands of surface weather maps. This type of analysis is not feasible to repeat for even a single climate model run or even multiple reanalysis products. A second, more recent study [2] followed a similar approach for extreme precipitation and extreme wind events. They approached the problem by defining a 6 × 6-degree area of influence around a meteorological feature, such as a front or cyclone, and tagging all precipitation within as belonging to the feature. Because different processes may have varying spatial scales, this work will focus on detecting the driving process(es) on an individual grid point level.

In this work, the main objective is to create a set of algorithms that use model data to identify meteorological processes that cause individual extreme precipitation events. The focus is on three main processes: fronts, convection, and extratropical cyclones.

### 2.1. Frontal Identification

Objective frontal analysis dates back at least as far as the work of Renard [3] and debate on the precise definition of a front has persisted since the concept was introduced by Bjerknes [4]. Definitions include gradient in air temperature, surface humidity gradient, the leading edge of temperature advection, abrupt shift in wind direction, a change in the air's origin, and warm air side of the gradients in air temperature and low-level humidity [5]. Objective frontal detection schemes typically focus on a wind shift [6], a temperature gradient [7], a combination of moisture and temperature gradients [8–10], or a combination of temperature gradient and vorticity [11]. The vertical levels to which these schemes are applied depend on the application and vary from surface fields to around 850 hPa. Some schemes look at upper-level fronts (600 hPa) but they are invariably secondary to the analysis. The schemes of each type are broadly similar but tuned to the specific needs of those using them. The specifics of one scheme of each type will be discussed in greater detail below.

### 2.1.1. Wind Shifts

Simmonds [6] looks for winds to shift from the northwest quadrant to the southwest quadrant and for the change in meridional velocity to be greater than 2 m/s over a 6-h interval. This scheme is designed to look at southern hemisphere fronts, so the direction of the wind shifts would need to be reversed for application to the northern hemisphere. A grid point is flagged as a front at the end of a 6-h interval that meets these criteria. Adjacent frontal grid points are considered to be frontal objects and single unconnected frontal grid points are discarded. The front is then found to be the smoothed eastern edge of each frontal object. For each front, the angle (relative to the meridian), the center of gravity, length, and intensity is recorded. The intensity is the sum of all the changes in meridional velocity along the length of the front, normalized for different spatial and temporal scales. This scheme cannot detect stationary or very slow-moving fronts, both of which can create flash flood events [12]. Although this scheme is useful in detecting and tracking mobile fronts, especially over water, and possesses a natural grid-point-based measure of intensity, its emphasis on 10 m winds and inability to detect slow-moving or stationary fronts are severe drawbacks.

### 2.1.2. Temperature Gradient Only

Mills [7] uses the gradient of air temperature at 850 hPa to measure frontal strength to connect strong pre and post-frontal winds to major fire events in Australia. The author thought that if the temperature gradient was strong at 850 hPa, then that indicated a deeper tropospheric structure and strong associated winds. To connect this to precipitation, one would expect a strong thermal gradient and winds to create strong advection, generating

ascent and precipitation (if the conditions are moist enough). This work differs significantly from many other frontal analyses in that it does not attempt to locate a front, although it can be adapted for that purpose [13]. Instead, the goal is to find the area the front impacts with strong winds and measure the intensity of this impact. This is a very useful, impact-driven, frontal detection scheme. The primary limitation is that it cannot discern fronts arising from a sharp moisture gradient between two air masses, as commonly arises from the "Dry-line" in the central U.S.

### 2.1.3. Thermal Frontal Parameter

Catto [8] uses a thermal frontal parameter (TFP) defined by Renard [3] to identify fronts to link them to precipitation events. This TFP is defined, for some scalar variable $\tau$, as the derivative of the magnitude of the gradient of $\tau$ in the direction of the gradient of $\tau$ (Equation (1)).

$$TFP = -\nabla|\nabla\tau| \cdot \frac{\nabla\tau}{|\nabla\tau|} \tag{1}$$

In [8], $\tau$ is the wet bulb potential temperature at 850 hPa. Points are identified where both the TFP is negative and the gradient of the TFP is zero, then those points are linked together to form fronts. This is undertaken on a very coarse grid (2.5° resolution) to reduce the influence that numerical noise can have on the detection of the location of a front [6]. To associate precipitation (also on a 2.5° resolution) with a front, the grid box experiencing the precipitation and the surrounding eight grid boxes are searched for an instance of a front at the beginning and end of the 6 h of precipitation accumulation. If a front is found, then the precipitation event is associated with that front. Catto [8] notes that misallocation of precipitation is possible with all automated methods due to the area of influence of a front being substantially larger than the front itself. This method is also applied to daily precipitation data, in which the chance of misallocation is slightly higher; if a short-lived rainstorm occurs when no front is present, but a front passes through earlier or later in the day, the precipitation would be incorrectly flagged as frontal.

### 2.1.4. Vorticity and Temperature Gradient

Parfitt [11] develops a two-variable method for frontal detection. This method combines a thermal variable (the magnitude of the temperature gradient) with the relative vorticity seen in equation form below (Equation (2)).

$$F = \eta_p|\nabla(T_p)| \tag{2}$$

This quantity is divided by the Coriolis parameter and a baseline temperature gradient of $\frac{0.45 \text{ K}}{100 \text{ km}}$ to normalize the metric. Grid points with values exceeding one at 900 hPa are considered frontal. This threshold must be found via case study analysis at each pressure level to be considered, which is a substantial drawback. Additionally, the authors only test their metric over oceans, an ideal place for automated detection because it completely removes topography. This is not an option afforded to the analysis of fronts appearing over land.

### 2.2. Convective Identification

Thunderstorms and intense convection can often give rise to heavy and localized precipitation. Dowdy [2] uses a network of ground-based lightning detectors (World Wide Lightning Location Network: WWLLN). Grid cells with at least two lightning strikes during the 6-h precipitation time period of the study [2] are considered to be convective. This threshold indicates the presence of a deep convective storm co-located with the precipitation.

Synoptic cloud observations are used to distinguish convective precipitation from stratiform precipitation by Berg [14]. The authors classify precipitation as convective if observations find cumulus or cumulonimbus clouds during the period of precipitation.

In their large-scale manual identification of meteorological causes of extreme precipitation, Kunkel [1] has two classes of convective events, Air Mass Convection (AMC) and Mesoscale Convective Systems (MCS). Each type was characterized by a convectively unstable vertical temperature profile. An event was classified as AMC if the precipitation was isolated (existed at an isolated grid cell or pair of cells). AMC events were also typically found in warm areas and at times of the year. The MCS types often needed to be separated from their frontal category because MCSs often spawn along frontal boundaries before separating. These MCSs are characterized by moderate southerly winds, sometimes lacking anomalously warm temperatures. Even so, MCS was often assigned as the category if no other category was appropriate.

The first two schemes rely on observational networks, which limit the applicability of techniques to reanalysis data. Each of these schemes seeks signs of existing vertical instability associated with an extreme event to classify the event as convective.

### 2.3. Vortical Identification

Vortical features are often investigated through the study of extratropical cyclones. Many detection schemes have been implemented to find and track these features, despite no universally accepted definition [15]. These tracking schemes usually use one or more of these four variables: mean sea level pressure (MSLP) [16–18], upper-level vorticity [17–19], lower level vorticity [20–22], and geopotential height at 850 hPa [20–22]. These methods use differing levels of terrain filtering depending on their geographic area of study or their method's reliability over topography. As with frontal identification, the misallocation of precipitation is possible by these automated methods due to the area of influence of an extratropical cyclone being substantially larger than the cyclone itself.

## 3. Methods

### 3.1. Data

This study uses the fifth version of the European Center for Medium-Range Weather Forecasting's (ECMWF) atmospheric reanalysis (ERA5) [23]. ERA5 precipitation was chosen for performing best in the extratropics and generally correctly identifying the location and spatial pattern of extreme precipitation [24]. The data perform worse in summer than in winter, but the errors are still relatively small. For this purpose, the location of extreme precipitation is more important than the exact amount. All ERA5 data used in this study were retrieved from the Climate Data Store (CDS) using their application programming interface (API). This interface allows us to regrid this dataset (from its natively stored quarter degree and hourly resolution) to one-degree resolution in the horizontal and three hours resolution in time. This regridding is undertaken by resampling the data, which is convenient because data exists at each desired point in space and time. It should be noted that this type of regridding does not conserve the total amount of precipitation and could weaken the thermal and vortical gradients used to identify certain processes. Because we focus on only extreme events, conserving the total precipitation is unlikely to have a large effect. These resolutions are attainable by regridding and/or data being readily available at these or finer resolutions from many GCMs/reanalysis outputs, making them a good starting point for this analysis. This work defines an extreme event to be the top 5% of nonzero 3-h precipitation accumulation at each grid cell. This percentile method of defining extremes is based on the work of the Intergovernmental Panel on Climate Change [25]. A resolution of 3 h is short enough that one can match instantaneous measurements of process strength to the accumulated precipitation. The period from 1980–2010 is used to diagnose the climatology of PEx drivers in the current climate. This period was chosen to end in 2010 to align with the period of the Kunkel [1] study. Beginning in 1980 allows the period to match the approximately 30-year length of other definitions of current climate [26,27].

*3.2. Metrics for the Identification of Processes*

For each of the three processes of interest (fronts, convection, and cyclones), a simple and easy-to-calculate metric to measure its strength is created. These metrics take inspiration from previous work and attempt to build on their commonalities. A significant difference between this and the approaches outlined above is the desire to quantify the contribution of each process at the grid point level. This desire motivated the choice of metrics.

### 3.2.1. Frontal Intensity

To identify the presence and quantify the strength of fronts associated with the precipitation extreme event, the absolute value of the horizontal gradient of equivalent potential temperature at 850 hPa (Equation (3)) is used.

$$Frontal\ Metric = |\nabla_p \theta_e| \tag{3}$$

This metric accounts for both temperature and moisture gradients as precipitation can arise from sharp gradients of both kinds between air masses [3]. Because PEx can arise some distance from a frontal surface and because the data have 3-h time steps, the frontal metric score is the largest among all grid points surrounding the PEx at the beginning or end of the 3-h accumulation period.

### 3.2.2. Convective Intensity

Convection arises from static instability in the atmosphere and can generate very high precipitation rates over a relatively small area. The change in Convective Available Potential Energy (CAPE; Equation (4)) during the precipitation event is used to measure the severity of the convection. This metric uses CAPE measured for an air parcel lifted from the surface.

$$Convection\ Metric = \frac{\partial}{\partial t} CAPE \tag{4}$$

This metric measures the change in the stability of the column to convective motions during the event. Only negative values (decreases in CAPE) are considered an indication of convective activity. This was inspired by discussions on the nature of convection and CAPE in the context of statistical equilibrium by [28]. In particular, the idea is that the energy available for convection is proportional to the amount of CAPE that is present locally. It is also in line with the three schemes discussed previously in that it focuses on a sign of vertical instability, in this case, CAPE.

### 3.2.3. Vortical Intensity

The vorticity advection at 500 hPa (Equation (5)) is used to assess the impact of cyclones and strong vortical features.

$$Vorticity\ Metric = -U \cdot \nabla_p \zeta \tag{5}$$

Positive values of this metric are associated with increased upper-level vorticity. This could be because of the presence of a midlatitude cyclone. Alternatively, a less notable vortical feature may be present, which could still drive ascending motion through the column during the precipitation extreme. Similar to the frontal metric, the largest score from among neighboring grid points and from between the beginning and end of the precipitation accumulation period is chosen.

### 3.2.4. Scaling the Intensities

Because each of these metrics has different units, comparing their relative importance to a particular event carries some challenges. Several approaches to make the metrics inter-comparable were considered, with the primary one being placing each metric on a scale relative to the usual strength of the metric in a particular area. For this approach, a

0–100 scale based on the local percentile rank of each metric was created. Above-average frontal metric scores, negative convection metric scores, and positive vorticity metric scores are retained for each grid point. Other scores are set to 0 because they are judged to be either the wrong sign to directly generate precipitation (in the case of convection and vorticity) or unremarkable (in the case of below-average frontal scores). A below-average thermal gradient (including non-precipitating time periods) was judged unlikely to be the direct cause of an extreme precipitation event. The retained scores are then ranked on a scale from 1–100 at each grid point.

The relative scales are created based on every value of the field in the data. This purposefully includes times with non-extreme, and even zero, precipitation. This choice allows the separation of the rarity of each process from whether or not large amounts of moisture were present. When regional and seasonal statistics are discussed in the following sections, the events considered are every three hours with precipitation accumulation greater than the local 95th percentile of those periods with nonzero precipitation per the guidance of the (IPCC) [25].

### 3.3. Self-Organizing Map Regionalization

A self-organizing map (SOM) is used to identify regions that share the same mean precipitation seasonality [29]. This is very useful because it allows for the grouping of a set of extreme events (those occurring within a particular region). The seasonality of the processes driving these groups of events will be examined and compared to previous literature. The raw output from the SOM is shown in Figure 1, as well as its isolated area count (the median number of isolated areas comprising a single region: IAC), minor areas fraction (the median fraction of a region's area that is not contained in the largest isolated area: MAF), and compactness ratio (the median ratio of the square root of a region's area to its perimeter) which all compare well to the scores in the results of Swenson [29]. Additionally, this set of regions passes the regional extremes ratio (RER) threshold of 20% set out in Swenson [29] using this paper's three-hour definition of a precipitation extreme. Because of the lower resolution used in this work, the created maps use fewer regions than in Swenson [29]. This results in Florida (FL) not being separated from the New Mexico/Texas border region naturally. This necessitates the use of the automatic intervention to separate large enough isolated areas grouped by the SOM discussed in Swenson [29] to separate FL into a 7th region. Further discussion of the link between the seasonality of precipitation in FL and the New Mexico/Texas border region can be found in Swenson [29]. The borders between regions from the analysis are removed to reduce the uncertainty in the seasonal cycle in each region. The final regional map is shown in Figure 2. This Figure 2 does not display the scores because IAC and MAF are one and zero, respectively, because of the processing, while the compactness ratio increases. After processing, the RER is the only criterion likely to be negatively affected and the threshold is still passed for the processed regions.

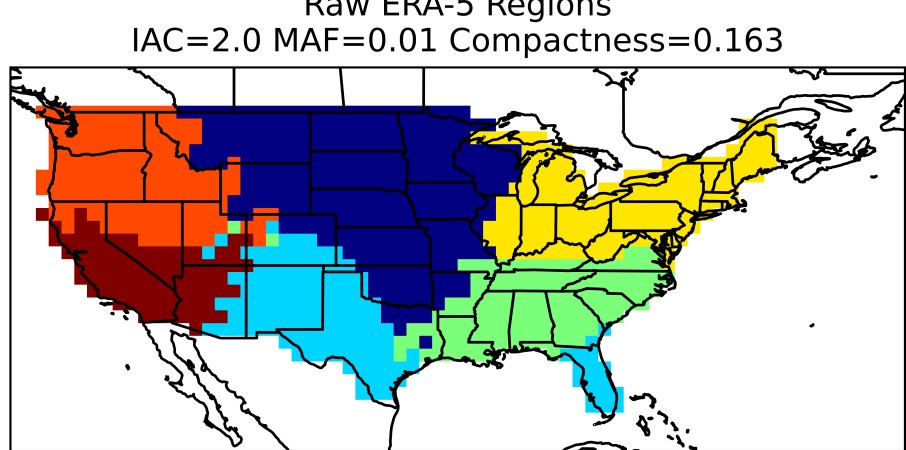

**Figure 1.** The raw output of the SOM trained on the normalized long-term daily mean of the cube root of precipitation at each grid point as in Swenson [29]. IAC and MAF refer to the map's median isolated area count and minor area fraction, respectively.

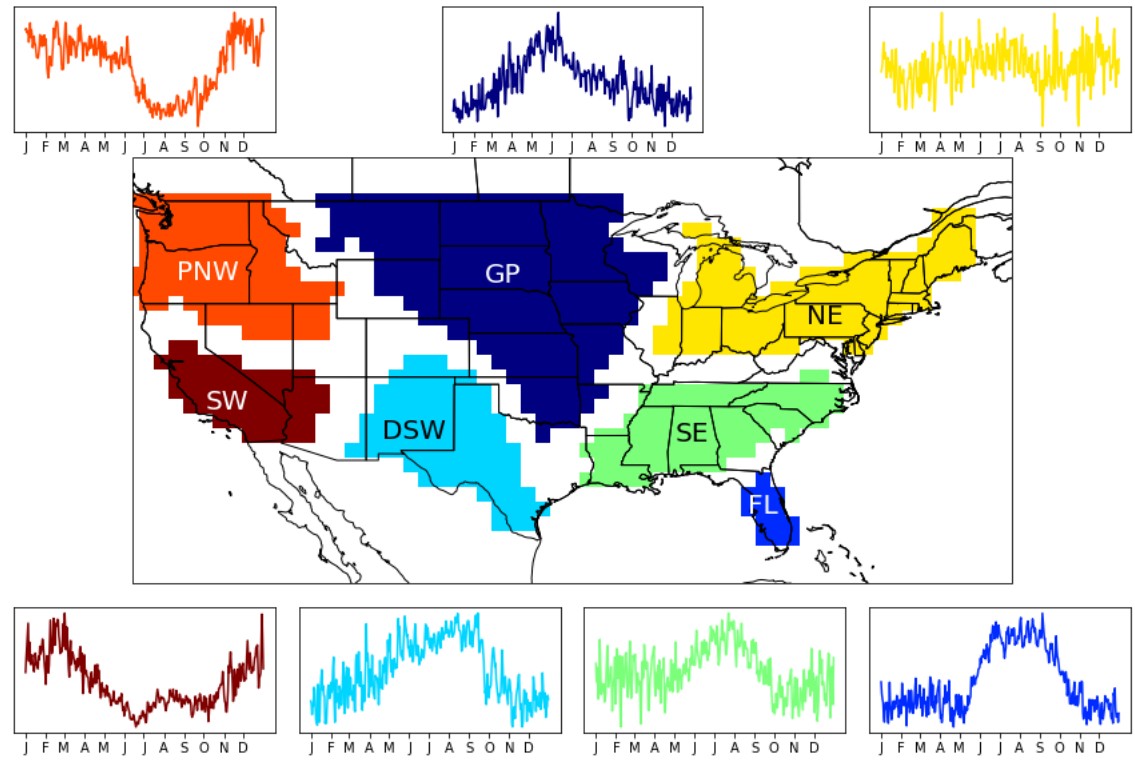

**Figure 2.** The seven regions of a similar annual cycle by which our results may be grouped. These regions result from the approach in [29] for ERA5 daily precipitation data. Abbreviations for the seven regions are as follows: Great Plains (GP), Florida (FL), Desert Southwest (DSW), Southeast (SE), Northeast (NE), Pacific Northwest (PNW), and Southwest (SW). The mean annual cycle of precipitation averaged over all grid points within each region is displayed in the subplot matching the color of the region. The *y*-axis of these subplots is the precipitation amount normalized to the region, with the limits chosen separately to span the normalized data for that region. The *x*-axis is the day of the year with tick marks on the first day of every month.

## 4. Results

### 4.1. Multivariate Metric Distributions

The simplest method to investigate the processes, as described previously, is to examine the various bivariate distributions of the metrics. An example of this is shown in

Figure 3. This example displays data from the Southeast region (Figure 2) during summertime extreme events. There, no extreme events with a score between 5 and 50 in the convective metric (upper left panel) can be seen. The overwhelming majority of the events in this 0–4 bin are events with a value of zero. This indicates that, for these extreme events, convection is either above-average strength or not existent. This is in contrast with the other two metrics shown in this figure, each of which shows a relatively flat distribution from 5–100 with the most frequent score between 0 and 4. The bivariate distributions show that convection is about as likely to be "on" in the presence of a front as it is in the presence of vorticity (comparing the distributions of convection vs. frontal to convection vs. vorticity). Looking at the center bottom panel, events with both a nonzero frontal and nonzero vortical score tend to have a higher vortical score than the frontal score.

In contrast, during wintertime, the mix of processes (Figure 4) associated with extreme precipitation is much different. Far fewer events are associated with high convection scores, but the gap in the convection score distribution remains (upper left panel of Figure 4). The nonzero scores in the vorticity and frontal metrics are no longer flat, but have many more events at higher scores. The quadrant where Vorticity and Fronts are both relatively weak is the most sparsely populated with extreme events (middle right panel of Figure 4).

As a comparison, let's look at a region that experiences most of its precipitation during wintertime, such as the Pacific Northwest (orange in Figure 2). In the summertime (Figure 5), the distributions are very comparable to those of the southeast (Figure 3), with an increased frequency of high scores in the vorticity and frontal metrics. However, in wintertime, there is a decrease in the relative strengths of the frontal metric scores compared to the vorticity metric scores (Figure 6) that is not seen in the southeast. This is somewhat surprising because fronts are discussed regularly in relation to precipitation in this region. However, previous studies disagree on whether the primary driver of winter extreme precipitation here is extratropical cyclones [1] or fronts [2]. Although the two processes are interrelated (extratropical cyclones commonly develop strong fronts [4]), this study finds unusually high levels of vorticity advection (which points to cyclonic activity) more commonly associated with wintertime extreme precipitation than strong thermal gradients in the Pacific Northwest.

### 4.2. Categorical Comparison

For the three metrics, eight categories are created: Front, Vorticity, Convection, Front and Vorticity, Front and Convection, Vorticity and Convection, All, and None. Only about 1% of events fall into the "None" category. This is beneficial in that these metrics capture most extremes, though this can also be interpreted as a sign that the metrics are too broad. A process is excluded as a factor for an event if it has either a score of 0 or a score that is 20 less than the process with the highest score. Events fall into the "None" category if all three metrics have a score of zero and into the "All" category if each metric is within 20 points of the other two. This 20-point threshold is a subjective choice, which was made to balance the desire to find the most important process and leave some room for the possibility of mixed-type events.

To examine the geographic and seasonal variations in the relative strengths and frequencies of each process associated with extreme precipitation, the percentage of events that fall into each category is plotted for each region during the entire year, the summer, and the winter (Figures 7–9, respectively). The most noticeable trend here is the increased fraction of events that are in the Convection category during summer for all regions. This is true for regions with strong precipitation seasonality (the Pacific Northwest, in orange), as well as a region with a flatter seasonal cycle (the Northeast, in yellow). In the annual plot (Figure 7), four regions have Convection as the category most associated with extreme precipitation: the Great Plains, Desert Southwest, Southeast, and Florida. The two west coast regions share Vorticity as their most frequent category. The Northeast is the only region with a mixed category (Vorticity and Frontal) as its single most common category during extreme precipitation. Unsurprisingly, Florida is the region with the largest gap

between its most and second-most common categories. Florida is also the region with the most extreme seasonal shift. It goes from 60% convection in summer to 35% vorticity in winter. Both of these are large percentages for a single category to occupy.

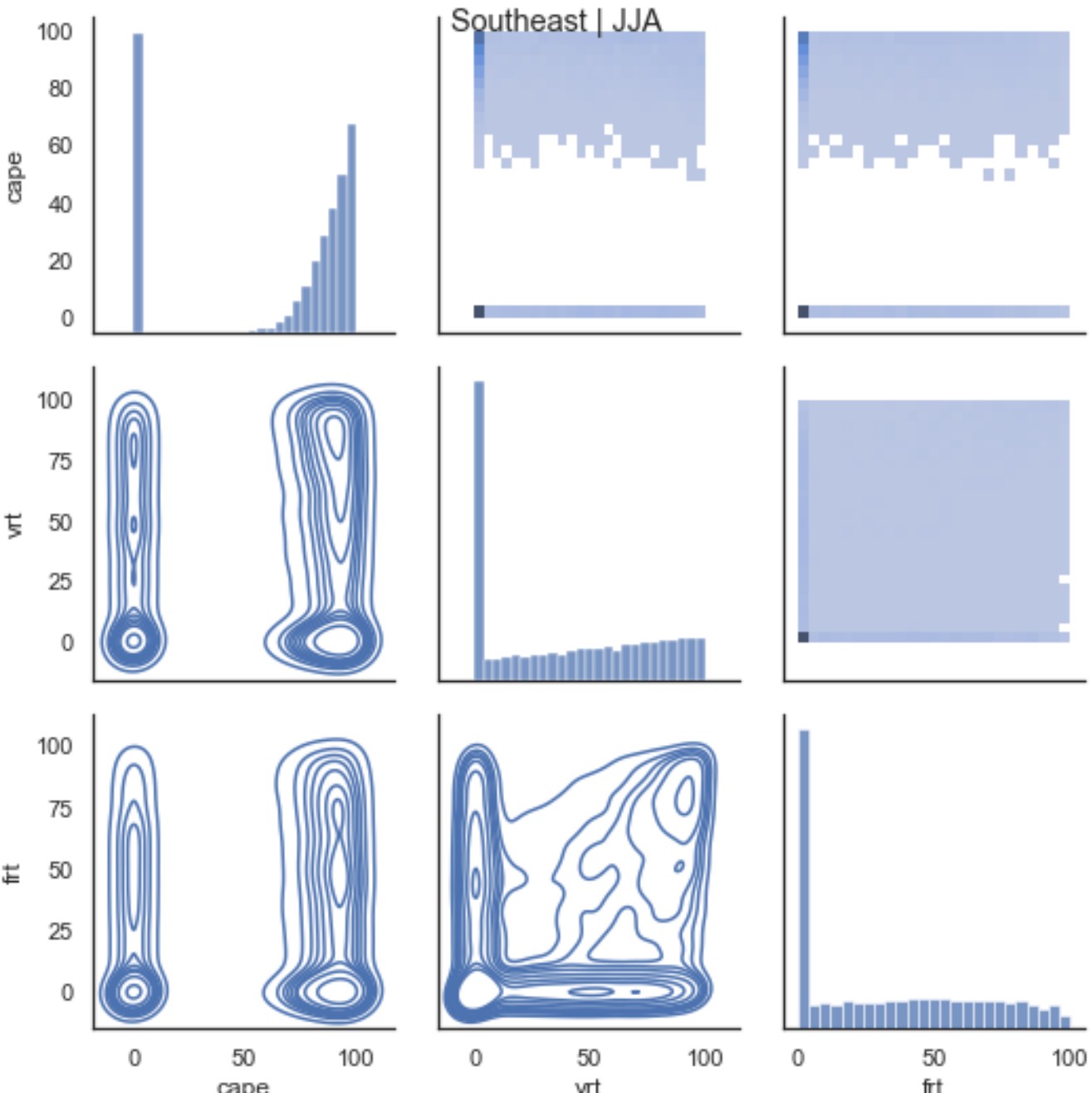

**Figure 3.** The plots along the diagonal are the univariate distribution of each of the three processes (Convection, Vorticity, and Frontal). The *x*-axis is the rank of the process (broken into 25 bins) and the *y*-axis is the number of events in that bin (the *y*-axis labels do not apply to the panels on the diagonal). The lower triangular panels show a kernel density plot of the two processes indicated on the corresponding *x* and *y* axis labels. The upper triangular panels show the same data as a bivariate histogram.

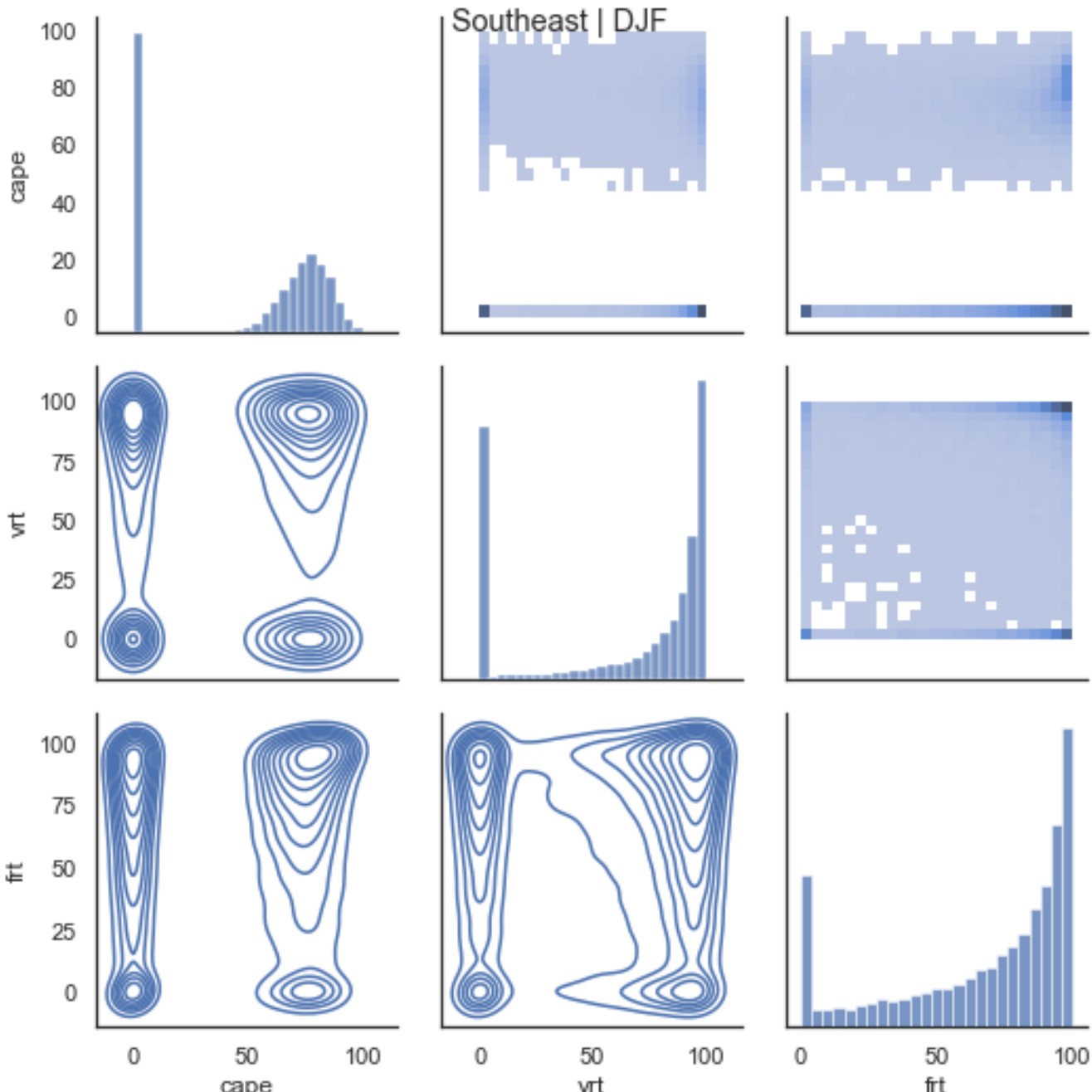

**Figure 4.** The plots along the diagonal are the univariate distribution of each of the three processes (Convection, Vorticity, and Frontal). The *x*-axis is the rank of the process and the *y*-axis is a density function over all events in the named region and season. The lower triangular panels show a kernel density plot of the two processes indicated on the corresponding *x* and *y* axis labels. The upper triangular panels show the same data as a bivariate histogram.

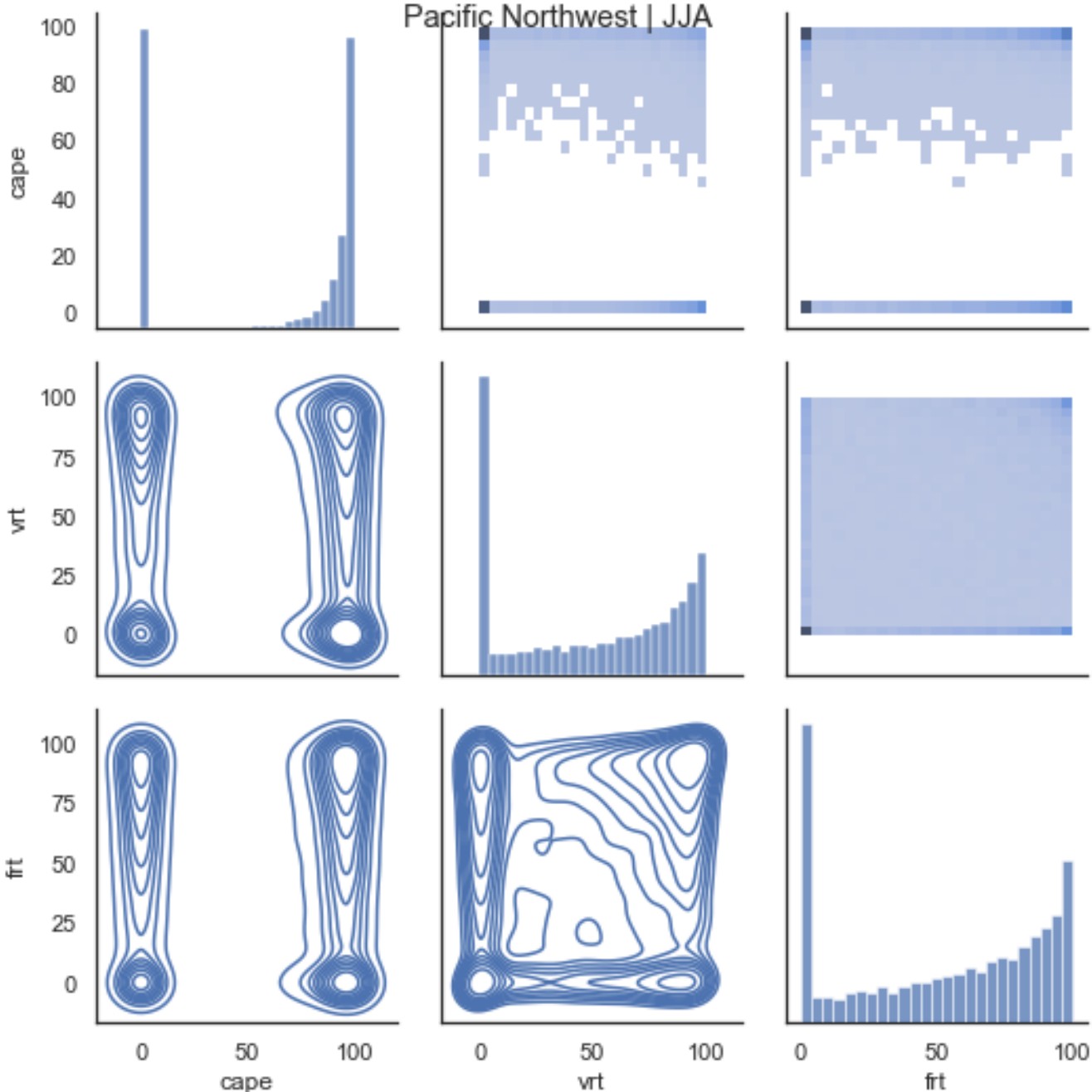

**Figure 5.** The plots along the diagonal are the univariate distribution of each of the three processes (Convection, Vorticity, and Frontal). The *x*-axis is the rank of the process and the *y*-axis is a density function over all events in the named region and season. The lower triangular panels show a kernel density plot of the two processes indicated on the corresponding *x* and *y* axis labels. The upper triangular panels show the same data as a bivariate histogram.

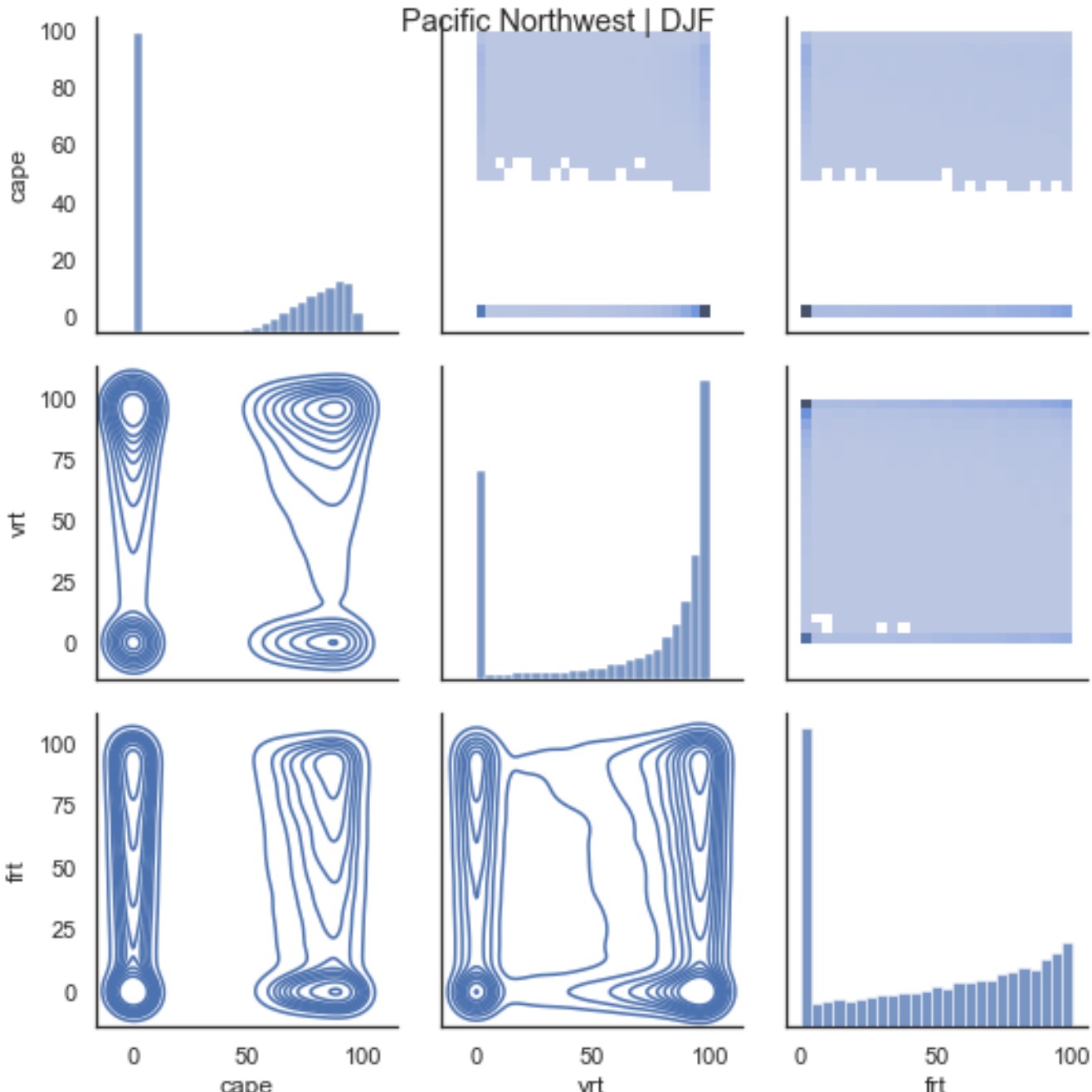

**Figure 6.** The plots along the diagonal are the univariate distribution of each of the three processes (Convection, Vorticity, and Frontal). The *x*-axis is the rank of the process and the *y*-axis is a density function over all events in the named region and season. The lower triangular panels show a kernel density plot of the two processes indicated on the corresponding *x* and *y* axis labels. The upper triangular panels show the same data as a bivariate histogram.

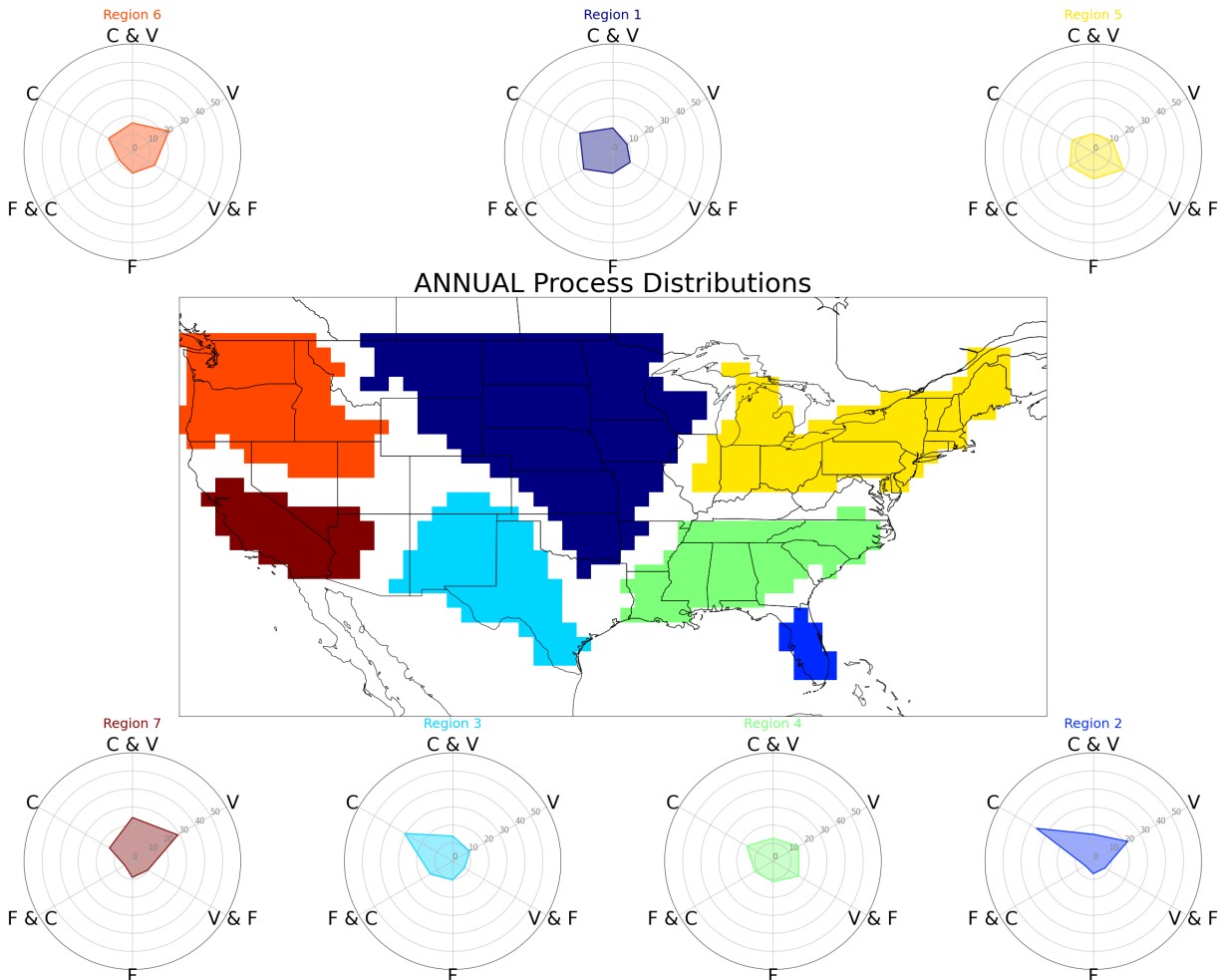

**Figure 7.** Central plot shows the regions over which extreme precipitation was aggregated. The distributions of the various categories for extreme precipitation within each region are shown around this central plot. Concentric rings are in increments of 10%. This includes every event in the record used, except those in the "ALL" and "None" categories. Those events were omitted to aid the visualization. "None" type events make up no more than 4% of the events of any combination of region and season.

### 4.2.1. Case Study—Front

As a way to check this analysis against previous work, several events from the previously mentioned Kunkel [1] paper are considered. This first event was placed into the frontal type based on the manual analysis of a limited set of meteorological fields. The results of this work's analysis are shown in Figure 10 at eight times during the day in question. The colored areas are where the precipitation threshold of 12.5 mm from Kunkel [1] is met. The comparison focuses on the largest connected extreme, running SW-NE from Ohio to Maine. In Figure 10, the categories that include fronts are prevalent throughout the breadth and duration of this event, which shows good agreement with the previous study. Figure 10 also shows definite changes in the mixtures of processes that are featured throughout the event in time. There is a shift in the secondary process whereby the Front and Convection category is present in the western part of the event early in the day and the eastern part of the event later in the day. This is coupled with the reverse trend in the Vorticity and Front category. Convection and Vorticity appear together, with less notable thermal gradients, in isolated pockets that fall mostly around the edges of the precipitation extreme. This case study largely agrees with the finding that this event is "Frontal", but it also reveals that this front is being detected by all three metrics. Indeed, no event that was

examined in this way was a truly "pure" case. The identified categories are quite varied in space and time. Some grid points belong to five different categories during the 24 h plotted in Figure 10. Notably, this precipitation event occurs during the last day of Hurricane Katrina's life cycle. The then-tropical-depression Katrina is near the southwestern edge of our precipitation event during this day.

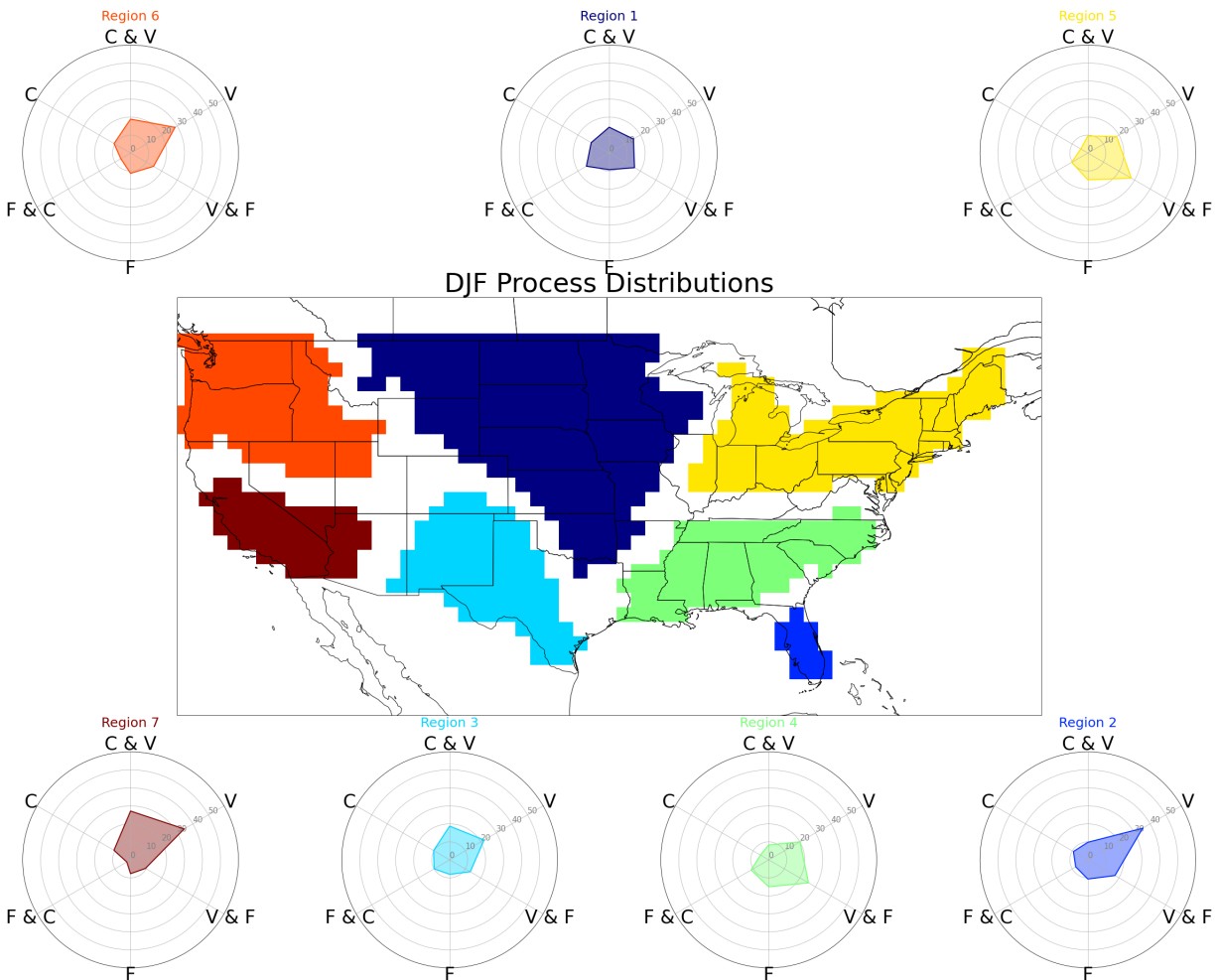

**Figure 8.** Same as Figure 7 but for only the winter months (DJF).

### 4.2.2. Case Study—Extratropical Cyclone

Figure 11 shows how the processes driving the extreme precipitation event of 17 June 1996 change throughout the course of the day. The focus will be on the area from Kansas up to southern Michigan, but it is briefly worth noting the other large area of strong precipitation east of Florida. This area lies in the path of tropical storm Arthur and is dominated by the vorticity and convection metrics. The event in the area of focus was classified as being primarily driven by an extratropical cyclone in Kunkel [1]. The metrics presented in this paper do not find unusually strong vorticity advection, compared to the strength of the thermal gradients for much of this event. The only place strong vorticity advection shows up is in the southern "tail" of the event and only around hour 12. For the rest of the event, the frontal and convective metrics show up much more strongly, and are often mixed. This event underlines both the ability for convection to be embedded in, or near, frontal features, and the difficulty of teasing apart the influences of extratropical cyclones and associated fronts. It can again be seen that all three metrics respond to this one event and generate varied results in space and time. It is interesting to note the way the categories with the frontal metric active begin as dominant over the region, while categories with convection active grow to dominate the area later.

### 4.2.3. Case Study—Front and Tropical Cyclone

The final case study (Figure 12) contains two separate events and event types, as classified by Kunkel [1]. The first event is tropical storm Erin creating precipitation over Oklahoma. This storm moves through north Texas into Oklahoma during the day of this precipitation event. In the metrics, this event shows up very strongly in terms of local thermal gradients, with the yellow, green, and cyan of the frontal categories well represented. This event looks very different than the area associated with tropical storm Arthur in the previous case (Figure 11). That event had almost no grid points that fell into any of the three frontal categories, whereas this one is dominated by them. Both events are associated with tropical storms, Arthur in the previous example and Erin in the current example. The primary differences are in the stage of development and the presence of land or water underneath the storm, which causes them to appear very differently in these metrics.

The other event was classified as a front by Kunkel [1]. Though the frontal categories are very present here, there are also a few areas that fall outside their scope, primarily in the Convection and Vorticity category. There are also significant areas where all three metrics are closely matched in score. Though a front is present, this event serves as an example of the mixed nature of extreme precipitation events, which makes it difficult to distinguish the primary cause from these metrics. As in the previous two examples, all three metrics return high scores during this one event, generating varied results in space and time. For example, some areas along the southern edge of the front take up three different categories in the first 12 h of the event.

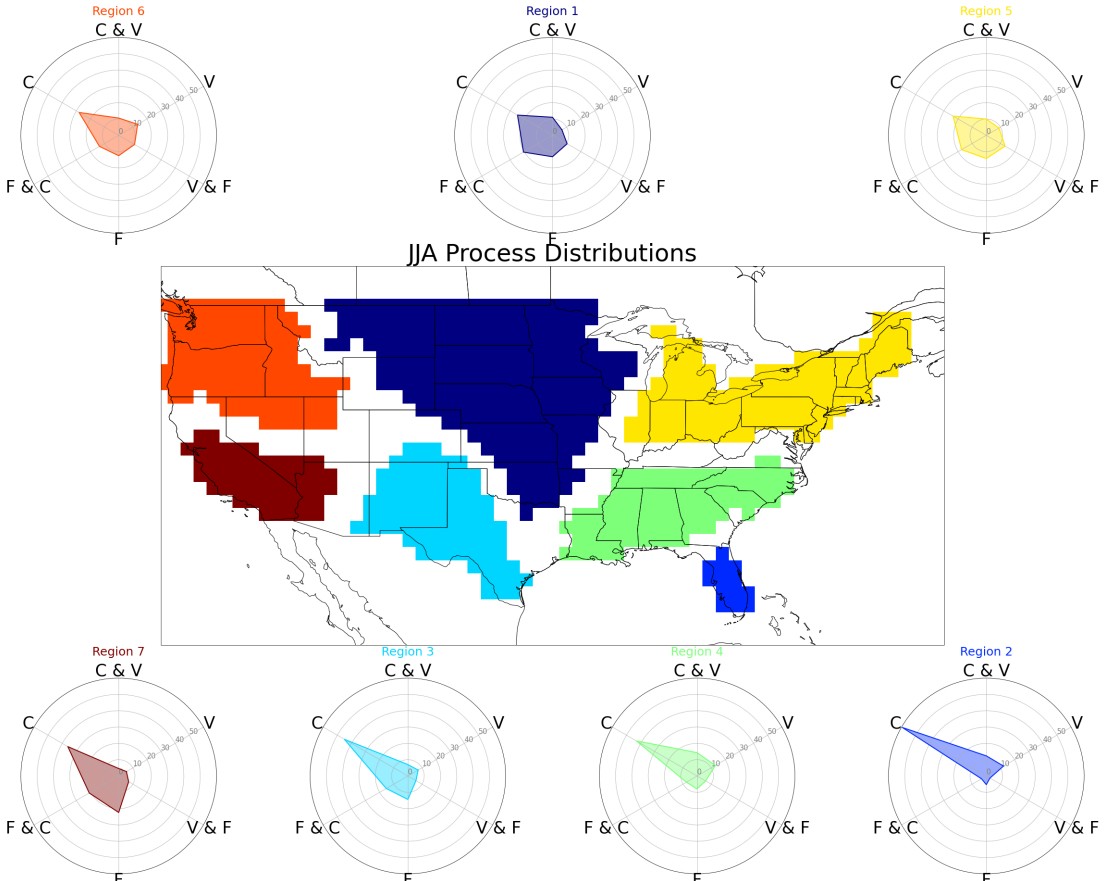

**Figure 9.** Same as Figure 7 but for only the summer months (JJA).

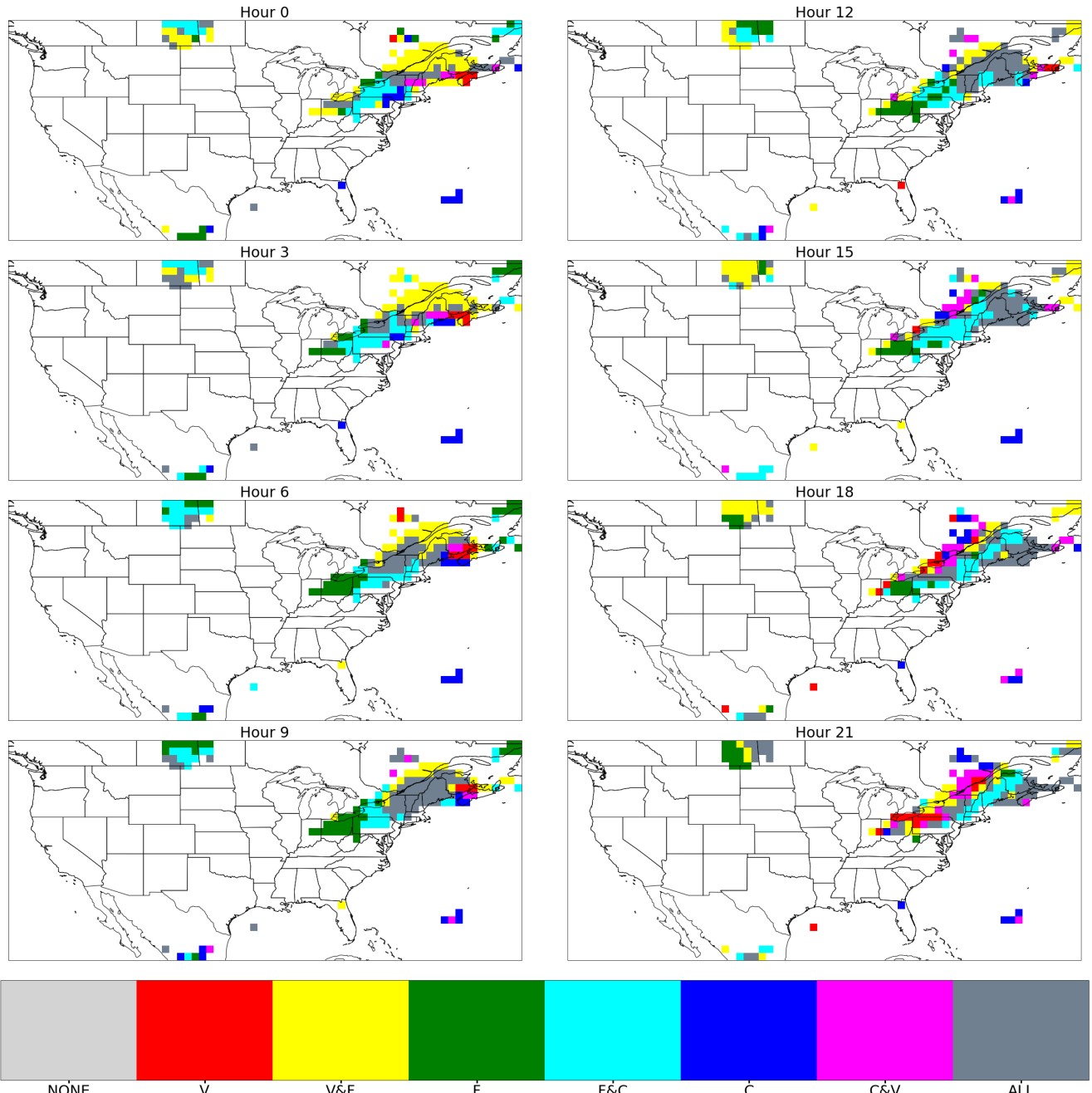

**Figure 10.** Plot of the process category for each grid point with precipitation $> 12.5 \frac{\text{mm}}{\text{day}}$ eight times during the day of 31 August 2005. The categories are labeled as follows: V—Vorticity, F—Frontal, C—Convection. The area of interest is the largest connected area of colored dots, classified as a frontal extreme by Kunkel [1]. All times are UTC.

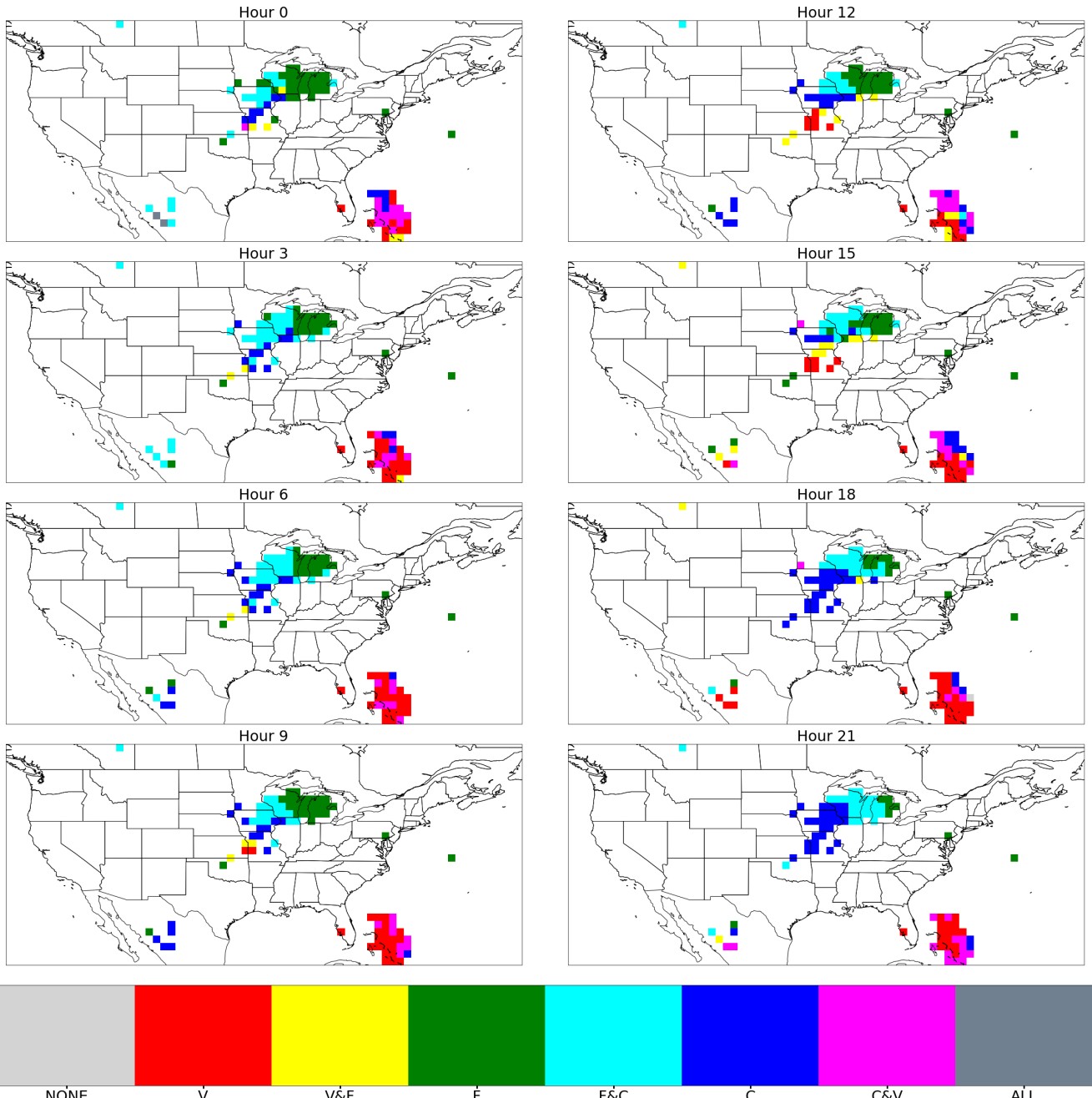

**Figure 11.** Plot of the process category for each grid point with precipitation $> 12.5 \frac{\text{mm}}{\text{day}}$ eight times during the day of 17 June 1996. The categories are labeled as follows: V—Vorticity, F—Frontal, C—Convection. The area of interest is the connected area of colored dots surrounding Wisconsin, labeled an extratropical cyclone by Kunkel [1]. All times are UTC.

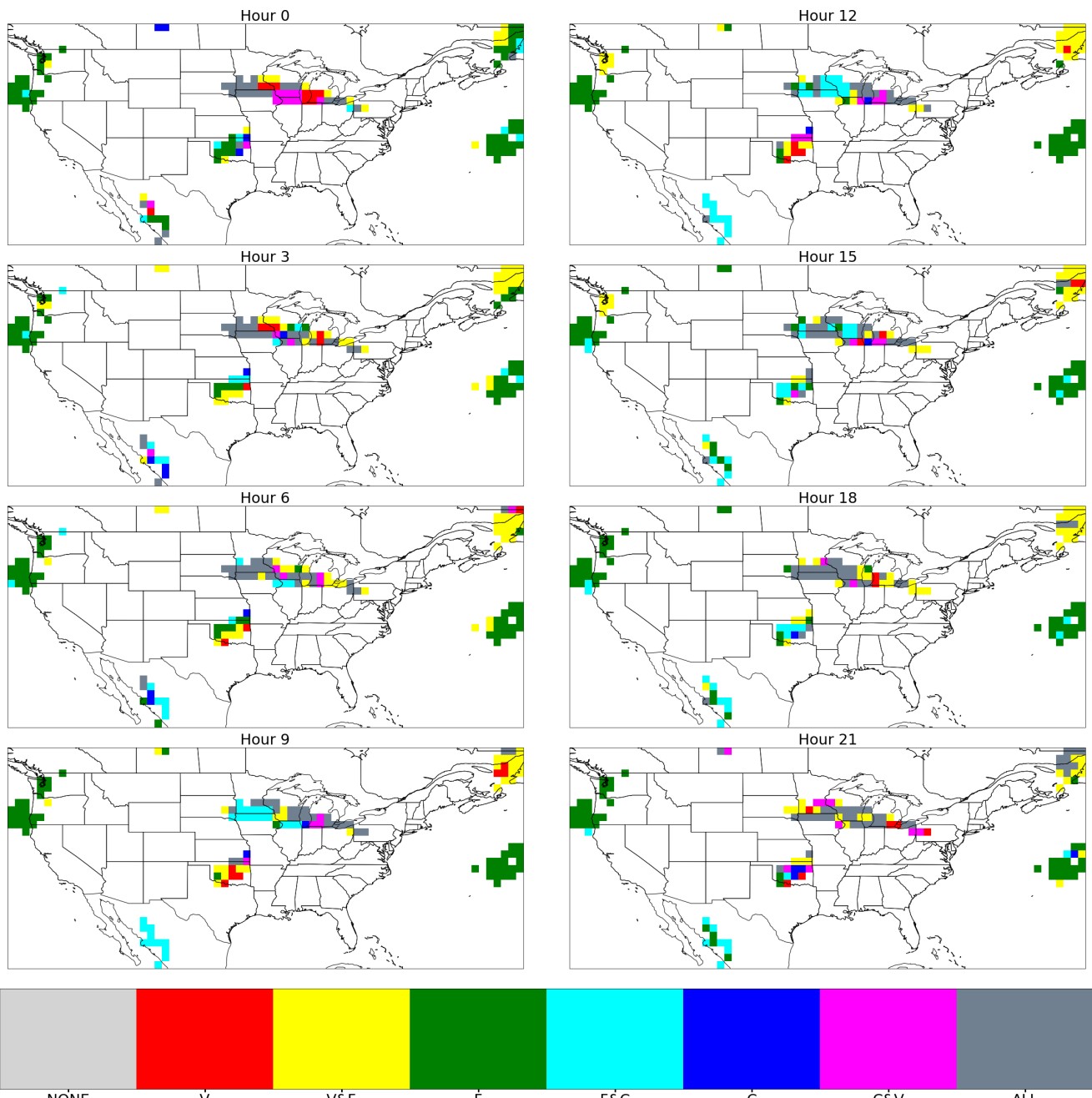

**Figure 12.** Plot of the process category for each grid point with precipitation $> 12.5 \frac{\text{mm}}{\text{day}}$ eight times during the day of 19 August 2007. The categories are labeled as follows: V—Vorticity, F—Frontal, C—Convection. The areas of interest are Oklahoma (classified as a tropical cyclone by Kunkel [1]) and the long line stretching from South Dakota to Pennsylvania (classified as a front by Kunkel [1]). All times are UTC.

## 5. Discussion

This work has presented an approach to identifying the process or processes responsible for an extreme precipitation event. The three processes were chosen to be commonly found processes present in the vast majority of events. This aspect was successful, as

only 1% of events were not captured by at least one of our metrics. The downside is that there is a large overlap between each of the processes, which can be seen clearly in the fractured nature of the case studies (Figures 10–12). This fracturing happens in both space (the mosaic-like appearance of the plots) and time (the rapid changing of categories during the course of each event at some grid points). Because of this overlap between the metrics, none of the analyzed cases fell into spatially or temporally consistent categories.

Some examples of how these metrics overlap include mesoscale convective systems (MCSs), and squall lines. MCSs are often born along frontal boundaries before separating [1]. These events would score strongly in both our convective and frontal metrics. Depending on the presence and position of a trough, the vortical metric could also score highly in some of these MCSs. Squall lines are a convective system even more closely tied to frontal activity. These types of events could contribute to the temporal inconsistencies by reaching an area just before the passage of a front [30]. Additionally, it should be noted that CAPE is simply a measure of the current state of the atmosphere and thus can be affected by processes that are not strictly convection. CAPE can be decreased without convection by the advection of less buoyant air into the column at low levels. It is also possible for CAPE to decrease due to the heating generated by precipitation whether or not the precipitation was generated by convection. This can lead to an increase in the convection metric without there being an actual increase in convection, which can contribute to the fractured nature of these case study events.

One of the ways fronts are sometimes identified is by observing shifts in the wind direction along the frontal boundary. This can lead to increased vorticity and vorticity advection in the vicinity of the front. A link between the frontal and vortical metrics also arises because both fronts (areas of strong thermal gradients) and troughs (areas with a largely cyclonic flow) often occur together. The frontal and vortical metrics are also both related to the detection of frontal cyclones. These systems will have areas of strong vorticity advection that overlap with frontal features, giving both metrics high scores. If, for instance, a squall line develops near a front in a frontal cyclone, all three of our metrics will score highly.

In some ways, these results are enlightening. During the case studies, very general trends (either changing in time or moving in space) in the importance of certain metrics to extreme events are observed. This interrelation between the processes does not permit the atmospheric process most associated with a particular precipitation extreme to be cleanly identified.

Climatologically, these metrics show increased strength of vorticity advection to wintertime extreme precipitation on the west coast (the Pacific Northwest and Southwest regions) relative to frontal strength (Figure 8). Though this is perhaps initially surprising, previous studies [1,2] have disagreed on how to disentangle extratropical cyclones from their associated fronts as the cause of extreme precipitation. This method finds a stronger relationship between the vortical field and precipitation extremes rather than the thermodynamic field. These metrics also show a much stronger shift towards convection in the summer (Figure 9) than found by Kunkel [1].

In the Northeast, the three frontal categories make up a higher proportion of events than the three vortical categories over the course of the year, but only narrowly (Figure 7). Kunkel [1] finds a more pronounced difference, wherein fronts are more common than extratropical cyclones by greater than a two to one margin. Dowdy [2] is in loose agreement with both the findings of this paper and the findings of Kunkel [1]. Dowdy [2] classifies the most frequent cause of extreme precipitation in this area as either associated with a mix of fronts and thunderstorms, or with a mix of cyclones, fronts, and thunderstorms.

In the Desert Southwest, a majority of events are convectively influenced. This is in agreement with the findings of Dowdy [2] and disagreement with Kunkel [1], who finds more than half of events to be frontal in nature. These direct comparisons are somewhat difficult given the different data, methods, and regional aggregations at play. However,

the disagreements between the previous studies and between this work and the previous studies highlight the uncertainty remaining in this kind of endeavor.

The FL region is most influenced by the convective metric annually. This agrees with the finding of Dowdy [2] that thunderstorms are the most frequent cause of extreme precipitation in FL. The Southeast and FL regions presented here are combined by Kunkel [1]'s work, which finds that more than half of all events are due to tropical cyclones and about a third of events are caused by fronts. Dowdy [2] finds a combination of fronts and thunderstorms to be the most common cause of extreme precipitation in the Southeast, which agrees with Figure 7 showing that region with a very even distribution of metrics represented.

In the Great Plains region, this work finds that the frontal metric is the most represented among the causes of extreme precipitation (Figure 7). This is in good agreement with both Dowdy [2] and Kunkel [1], which both find fronts to be the leading cause of extreme precipitation in this area. The seasonal analysis (Figures 8 and 9) also agrees with Kunkel [1] that fronts remain the primary driver throughout the year.

## 6. Conclusions

The three processes of focus are Convection, Vorticity Advection, and Fronts, which form an analogous approach to the work in [2]. Each process was given a metric that had a physical link to the mechanism used to create extreme precipitation: thermal gradients for fronts, vertical instability for convection, and upper-level vorticity advection for cyclones.

For convection, the metric is based on the amount of CAPE consumed during the event. For vorticity, the metric is based on the amount of positive vorticity advection. For fronts, the metric is based on the strength of the local thermal gradient. These metrics emphasize the strength of each process relative to the strengths found at each location. This avoids the creation of thresholds for each process that correspond to similar strength or importance between processes. The definition of these thresholds would be problematic because these processes are often linked, as demonstrated by the earlier discussion of extratropical cyclones and fronts. This work clearly shows that these physically relevant metrics are not strictly related to a single type of atmospheric process, and therefore do not separate extreme events cleanly. This shows that these physically reasonable metrics are unable to determine the main cause of extreme precipitation.

The metrics are created using simple calculations that can be done at the grid-point level to investigate the spatial change in process mix within an event (Figures 10–12). This comparison showed broad agreement with [1] when considering the influence of fronts. The metrics helped us identify areas where convection and vorticity each played the primary supporting role and how those areas changed during the course of each event. In the first and third events, the "front" was somewhat close to a tropical storm. The remnants of Hurricane Katrina were near the first case and tropical storm Erin was near the third. Disentangling a probable moisture source from the dynamic process is a challenge for any process attribution framework. Sometimes a choice needs to be made between the source of anomalously high moisture and the dynamic process responsible for the lifting and condensation. This emphasizes a takeaway from this work that synoptic-scale weather systems can create extreme precipitation through a variety of processes and that the primary cause cannot be identified from these physically based, grid-point-level metrics.

**Funding:** This research was funded by the U.S. Department of Energy, Office of Science, Office of Biological and Environmental Research program under Award DE-SC0016605 "A framework for improving analysis and modeling of Earth system and intersectoral dynamics at regional scales".

**Data Availability Statement:** The data used in this work comes from the fifth version of the European Center of Medium-range Weather Forecasting's (ECMWF) atmospheric reanalysis (ERA5). These and other data may be accessed using their application programming interface (API) or through the Climate Data Store website: https://cds.climate.copernicus.eu/#!/search?text=ERA5&type=dataset (accessed on 29 March 2023).

**Acknowledgments:** The author would like to thank Richard Grotjahn, Matthew Igel, Paul Ullrich, John C.H. Chiang, and Erwan Monier for the time they took to provide feedback during the duration of this project.

**Conflicts of Interest:** The authors declare no conflict of interest.

**Abbreviations**

The following abbreviations are used in this manuscript:

| | |
|---|---|
| AMC | Air Mass Convection |
| API | Application Programming Interface |
| CAPE | Convective Available Potential Energy |
| DSW | Desert Southwest |
| ECMWF | European Center of Medium-range Weather Forecasting |
| ERA5 | ECMWF Atmospheric Reanalysis 5 |
| FL | Florida |
| GP | Great Plains |
| MCS | Mesoscale Convective System |
| NE | Northeast |
| PEx | Precipitation Extreme |
| PNW | Pacific Northwest |
| SE | Southeast |
| SW | Southwest |

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
