# Peer review of "Can Simple Metrics Identify the Process(es) Driving Extreme Precipitation?"

_climate, doi:10.3390/cli11040088_

Round 1
Reviewer 1 Report
This is an interesting paper. I recommend the paper be published with minor revisions.
1. In the keyword, climate change should be considered.
2. The introduction/background could be reorganized based on the journal structure, please revised focusing on the background, motivation, objectives, and significance of the study. I would suggest to add a more detailed overview of these.
3. What’s new in your study, please specify.
4. Why the author selected 1980-2010 as the study period and what is the resolution of the datasets? Need more clear statement.
5. I would suggest to add a methodological flowchart in the method section.
6. Please mention some shortcomings in this research; uncertainties involved with data and methods.
Reviewer 2 Report
he manuscript describe a simple method to identify extreme events using some indicators. The paper is well written and describes exhaustively the topic, even if the procedure is not so automatic, because a threshold value was chosen in arbitrary way (line 268).
Before publish the manuscript, there are two main questions that, in my opinion, need to be clarified.
1) In section 3.1 the author affirms to use the ERA5 database. Even if there is the reference, it would be useful to specify the original resolution of the datasets and especially how it has been regridded. This is, in my opinion, a critical information because the regrid process can influence the physical parameters in inhomogeneous areas.
2) In section 3.2.4 the author rescales the data using a percentile scale. I like it, but I'd like to know if he tried to scale the data using arithmetic mean and standard deviation. Following the prescription of the Intergovernmental Panel on Climate Change (IPCC), percentiles is used to define extreme events. Allow me to suggest the author a reference to the IPCC, especially how he decides to use the 95th percentile as threshold (line 202).
Minor revisions:
- the measure units should be not in italic style (lines 55 and 290)
Round 2
Reviewer 2 Report
This is the second version of the manuscript. The author follows all the suggestions of the first review improving the paper. In my opionion, the manuscript can be accepted for the publication.